# Exploring the Relationship between Urban Youth Sentiment and the Built Environment Using Machine Learning and Weibo Comments

**DOI:** 10.3390/ijerph19084794

**Published:** 2022-04-15

**Authors:** Sutian Duan, Zhiyong Shen, Xiao Luo

**Affiliations:** Urban Mobility Institute, Tongji University, 4800 Cao’an Road, Shanghai 201804, China; duansutian@tongji.edu.cn

**Keywords:** sentiment, built environment, machine learning, Weibo comments, youth

## Abstract

As the relationship between the built environment and the sense of human experience becomes increasingly important, emotional geography has begun to focus on sentiments in space and time and improving the quality of urban construction from the perspective of public emotion and mental health. While youth is a powerful force in urban construction, there are no studies on the relationship between urban youth sentiments and the built environment. With the development of the Internet, social media has provided a large source of data for the metrics of youth sentiment. Based on data from more than 10,000 geolocated Sina Weibo comments posted over one week (from 19 to 25 July 2021) in Shanghai and using a machine learning algorithm for attention mechanism, this study calculates the sentiment label and sentiment intensity of each comment. Ten elements in five aspects were selected to assess the built environment at different scales and also to explore the correlations between built environment elements and sentiment intensity at different scales. The study finds that the overall sentiment of Shanghai youth tends to be negative. Sentiment intensity is significantly associated with most built environment elements at smaller scales. Urban youth have a higher proportion of both happy and sad sentiments, within which sad sentiments are more closely related to the built environment and are significantly related to all built environment elements. This study uses a deep learning algorithm to improve the accuracy of sentiment classification and confirms that the built environment has a great impact on sentiment. This research can help cities develop built environment optimization measures and policies to create positive emotional environments and enhance the well-being of urban youth.

## 1. Introduction

In recent years, with the continuous improvement of living standards, people have begun to pursue a better ecological environment and a happier life. In urban planning work, it is necessary to deal with the relationships involving various spatial interests, such as between urban construction space and ecological space, economic space and social space, and individual space and public space. In early urban construction, spatial evolution mainly expanded outward with construction, ignoring the sense of human experience in public space. At present, some megacities have entered the middle and late stages of urbanization, and urban construction has shifted toward a focus on internal renewal.

In Shanghai, for example, urban construction is mainly aimed at regenerating built-up land, which increases the need to evaluate and optimize the built-up environment. As the relationship between the built environment and people becomes increasingly important, greater emphasis should be placed on human-centered concepts in urban planning and design. In terms of the relationship between urban space and people, the degree to which the physical environment satisfies the physical and psychological needs of its users affects people’s sentiment and well-being.

Emotions, as a fundamental part of human behavior, fluctuate according to visual [1], environmental [2], and temporal [3] factors. There has been a lack of expression of emotion in urban construction and geography [4]. In the 1990s, Anderson and Smith proposed the concept of emotional geography, which focuses on the relationship between people, emotions, and space [5]. Emotional geography, which holds that emotion has temporal, spatial, and social characteristics, is an expansion of traditional Western geography [6]. The emergence of emotional geography marked the beginning of a transformation from a category of purely subjective mental notions to a broad social space. In terms of a research direction, emotional geography is based on geography, but it also involves many other fields, such as environmental science, biology, ecology, business, economics, sociology, and psychology; it is extremely interdisciplinary [7]. By studying medical environments, ecological environments, open spaces, and emotions, scholars have also realized that space is not a cold geometric; it also carries a wealth of people’s sentiments. In urban studies, attention to spatial emotions can help improve cities, enhance the quality of spaces, address urban socio-spatial inequalities, and enhance the well-being of residents.

Sentiment is an expression of emotion. With the rapid development of the Internet and the accelerated pace of life, social media has changed the way people share and collect information. For example, Twitter, Facebook, Weibo, and other platforms have gradually become the main channels for people to share their lives, collect information, and express their emotions. The rapid development of the Internet has produced what is called “big data”. In the study of urban issues, big data helps researchers to approach concepts from the perspective to the users, which is the key when focusing on human-centered concepts. At the same time, big data has also enriched basic information and analysis methods for planning research. In the past, to understand public feedback, statistical information and social research were formed mainly through questionnaires and interviews. The data obtained by these traditional methods are passive, static, and small-sample data, which have certain limitations. Big data, on the other hand, is relatively low-cost to obtain, and the data samples are large, dynamic, and sustainable. Using big data, it is easier to explore the relationship between people and the environment.

As the largest social media platform in China, Weibo has become a place for people to express their feelings and opinions; it has a wealth of user sentiment data [8]. In Shanghai, by the end of 2020, the permanent population was 24.8709 million, and the average age of the population was about 38. According to the “Weibo User Development Report in 2020”, those people born after the 1980s accounted for 96% of all active Weibo users (as shown in Figure 1). From the perspective of age composition, Weibo data can reflect the sentiment characteristics of most youth.

With the development of computer technology, techniques for mining emotions from text have emerged. Text sentiment analysis technology (TSAT) refers to the semantic orientation or polarity analysis of subjective attitudes and emotions in texts using natural language processing (NLP), statistical, or machine learning techniques. After continuous research and development, TSAT has been improved from a sentiment-dictionary-based method to a machine-learning-based method, and from supervised machine learning to deep learning. TSAT based on deep learning can be combined with spatial study, and good results have been obtained. However, current research is not yet sufficient to conduct fine-grain sentiment classification studies.

This research uses machine learning algorithms based on attention to analyze Weibo comments with fine-grain sentiment classification through data obtained from the Shanghai Weibo platform. By analyzing the distribution characteristics of urban youth sentiments and studying the relationship between sentiments and the built environment in Shanghai, this study enriches the research on emotion and space. From the perspective of sentiment, a non-traditional perspective of built environment evaluation is established to provide people-centered strategies for improving the quality of urban environment. Our study contributes mainly

To develop a novel and high-accuracy machine-learning-based method for analyzing youth sentiment in Shanghai.To identify the determinants of sentiment as well as its relationship with the built environment in Shanghai.Through the machine learning technology, which can be utilized as a sentiment feedback system for improvement of urban construction.

## 2. Literature Reviews

### 2.1. Research on the Relationship between Sentiment and Spatial Environment

Research on the relationship between sentiment and space dates back to the 1950s, when some scholars argued that space could cause emotions. In 1970s, Western scholars proposed concepts related to human-geographical emotions, such as geopathic episodes, sense of community, and sense of place [9]. With the continuing process of urbanization, studies on the relationship between cities and people from the perspective of emotions have gradually increased, such as traffic congestion and quality issues with facilities and services, which affect the construction of living environments and the development of cities. In 2008, ‘Emotion, Space and Society’, a special journal on emotional geography, began to be published in order to provide a broad platform for interdisciplinary research on the spatial and social aspects of emotion [10]. Many studies on emotion and space have emerged, providing a new perspective for the construction of the spatial environment.

From the perspective of research scale, research on the relationship between public sentiment and built environment has experienced a shift in focus from macro to micro. At the macro scale, some studies confirm that the built environment is accompanied by emotional characteristics in the development process. Dai et al. combined the emotional map with urban spatial structure, transportation networks, and land use and other built-environment information, and found that the spatial structure and spatial attributes of Shenzhen had certain emotional characteristics in the process of development [11]. Koenig conducted emotional analysis on the physiological level through data from wearables and found that the emotional changes of pedestrians are related to the spatial sequence changes of streets [12]. At a smaller scale, emotions take on different characteristics in different environmental spaces. Chen et al. used social network data to carry out perceptual evaluation of urban open space and found that positive emotions are mainly concentrated in entertainment, sports, and food service [13]. Cui et al. attempted to build a deep learning framework based on social media check-in data to evaluate and analyze spatial emotional perception, finding that the proportion of positive emotions showed an overall decrease with the increase of the distance from the city center [14]. Some scholars also use the spatial distribution of emotional characteristics to divide spatial groups and recognize urban functional areas. Cranshaw et al. used social media data to mine residents’ emotional data for dynamic identification of urban functional areas, and their results provide strong support for discovered clusters and reveal the distinctly characterized areas of the city [15]. In recent years, sentiment research has emerged in specific environments, such as hospitals, parks, schools, and stores. Rahim et al. constructed a sentiment analyzer and a medical environment classifier based on patient online reviews as a basis for healthcare service improvement [16]. Hannah et al. conducted text mining of reviews on Airbnb to explore the quality of indoor environments [17]. In a specific environment, it analyzes the sense of experience from the built environment from reviews in order to improve the built environment.

In short, people in different places carry out different activities and perform different behaviors, resulting in different emotions. People will affect the emotional airborne pattern of a place, and the built environment of a place will also affect people’s emotions. Research on the relationship between sentiment and spatial environment can help to discover urban space issues and improve positive sentiment [18].

### 2.2. Text Sentiment Analysis

Usually, there are several difficulties in sentiment recognition. These are as follows. First, each person’s expression of the same or similar sentiment is different, especially in Chinese [19,20,21]. An exclamation or even a punctuation change can cause a reversal of text sentiment, which leads to difficulties in text annotation. Secondly, it is difficult to classify sentiment, and sometimes it is difficult to precisely understand the real sentiment from a sentence (often need to combine with contextual information), which leads to difficulty in sentiment recognition [22].

At present, methods for text sentiment analysis mainly consist of the sentiment-dictionary-based method and the machine-learning-based method. The sentiment-dictionary-based method uses existing sentiment dictionaries for analysis so as to calculate the sentiment polarity (positive, negative, or neutral sentiment) and sentiment intensity of a text by counting and weighting the sentiment-related words that appear in the text [23]. On the other hand, the machine-learning-based method uses machine learning algorithms to train a classification model by training a dataset with labeled sentiment polarity. Then, the classification model predicts the sentiment to which the text belongs.

#### 2.2.1. The Sentiment-Dictionary-Based Method

Sentiment dictionaries are used as an efficient analytical tool in sentiment analysis work. The construction of a sentiment dictionary is one of the most important basic tasks in the field of text mining. An appropriate sentiment dictionary plays an important role in the process of sentiment analysis, and it is important to select the appropriate method to build a domain-specific sentiment dictionary. Research institutions have released sentiment dictionaries such as HowNet [24], NTUSD [25], SentiWordNet [26], and Q-WordNet [27]. Turney’s work is representative of results based on the sentiment dictionary, using phrases containing adjectival adverbs as the main basis for determining sentiment polarity [28]. Using the Chinese emotion dictionary compiled by Hongfei’s team at Dalian University of Technology, Chao divided emotions into the seven categories of “joy”, “good”, “anger”, “sorrow”, “fear”, “evil”, and “shock” to evaluate the implementation of community emergency management projects from a public perspective [29]. In addition, Zhu proposed a new method for extracting Chinese microblogs’ subjective sentences, which is based on a combination of lexicon and corpus [30]. The sentiment-dictionary-based method needs the support of sentiment dictionary and judgment rules. The richer the sentiment dictionary, the more accurate the sentiment analysis will be.

However, there is no universal and complete sentiment dictionary, and common sentiment dictionaries do not perform well in scenario-specific research. With the development of the Internet, new Internet vocabularies are emerging. How to add these new online vocabularies to the existing sentiment dictionaries and construct a sentiment dictionary with high coverage has become a hot topic, with more and more research devoted to the automatic construction of sentiment dictionaries.

There are two main approaches for automatic construction of sentiment dictionaries: the semantic-dictionary-based approach and the corpus-based approach [31]. Tan used the HowNet sentiment dictionary as a basis and added the commonly used web terms from current dictionaries to construct a microblog emotion dictionary [32]. The experimental results showed that the semantic-dictionary-based approach achieved better results. Bai et al. presented a novel model to build a sentiment dictionary using the Word2vec tool based on our semantic orientation pointwise similarity distance (SO-SD) model [33]. This method enhances the automatic construction ability of a sentiment dictionary. In general, the sentiment-dictionary-based method also uses artificial intelligence techniques to automatically build sentiment dictionaries with broader coverage.

#### 2.2.2. Machine-Learning-Based Method

Data-driven deep learning methods can greatly improve the extraction and recognition of textual information. At present, machine-learning-based text sentiment classification methods include supervised learning methods and semi-supervised learning methods, among which supervised learning methods are the most common. The supervised learning methods include naive Bayesian, KNN, SVM, and random forest, among others. Meral and Diri used plain Bayesian support vector machines for sentiment analysis of Turkish on Twitter [34]. Mohammad et al. described a state-of-the-art sentiment analysis system, based on a supervised statistical text classification that detects the sentiment of short informal textual messages such as tweets and SMS messages [35]. Xuehua et al. conducted an analysis of public opinion related to COVID-19 in China based on a potential Dirichlet allocation model and a random forest algorithm [36]. While supervised machine learning techniques require large amounts of labeled training data, some semi-supervised methods are also used in text sentiment classification because their performance is less dependent on test data.

Text sentiment classification models can be divided into shallow models and deep models [37,38,39]. For shallow learning models, naive Bayes was the first model to be used in text categorization tasks. With the development of integrated learning, XGBoost and LightGBM provide better predictive capabilities. These models played an important role in the pre-development of deep learning because they did not require excessive training data. However, feature engineering is a difficult task for such methods [40]. Before training the classifier, researchers must collect knowledge or experience to extract features from the original text. Shallow learning methods train the initial classifier based on various text features extracted from the original text. For small datasets, shallow learning models usually show better performance than deep learning models under the limitation of computational complexity.

For deep learning models, researchers have proceeded in three general directions: recurrent neural networks (RNNs), convolutional neural networks (CNNs), and attentional mechanisms [41,42].

A CNN is an image classification method that uses convolutional filters to extract image features. For text classification, text needs to be represented as a vector, similar to the image representation. Text features can then be filtered from multiple perspectives. The commonly used model is the TextCNN model, which was the first model introduced to solve the text classification problem [43]. Johnson et al. proposed a text classification CNN model based on two-view semi-supervised learning, which first uses unlabeled data to train the embedding of text regions and then uses labeled data [44]. This method has achieved good results in text classification. Since convolution requires many calculations, CNNs have begun to consider the allocation of computational resources, such as a deep pyramid convolutional neural network with a little more computational accuracy, which is increased by raising the network depth [45].

RNNs are widely used in text (emotion) recognition due to their ability to capture long-range dependencies through recursive computation. Different networks based on RNNs will have different effects on text (emotion) recognition. Many researchers are also refining their models to achieve the best recognition capability. Abdi et al. presented a deep-learning-based method, called RNSA; it employs an RNN that is composed of long short-term memory (LSTM) to take advantage of sequential processing and overcome several flaws in traditional methods, where order and information about a word are lost [46]. Wang et al. proposed a model for text classification called BiMPM, which encodes input sentences with a BiLSTM encoder. The encoded sentences are then matched in two directions, which brings improvement in accuracy [47].

In recent years, the attention mechanism has been continuously used to improve attention to different visual and textual information. In 2015, Bahdanaau et al. first proposed an attention mechanism that can be used for machine translation [48]. Thus, researchers started to explore text or emotion recognition based on attentional mechanisms. Considering the attention mechanism, Yang et al. introduced the hierarchical attention network to obtain better visualization by exploiting the highly informative components of text [49]. In the later proposed Transformer, the attention mechanism is continuously used to improve the attention of different visual information.

Through RNN and attention mechanisms, text classification algorithms can be adapted to emotion recognition by capturing the temporal relationships in sentences. However, text classification algorithms should be optimized for performance in emotion recognition. For text classification, the subject, predicate, object information, and time sequence relationship play the most important roles. For emotion recognition, adverbs and adjectives of different strengths are significant. Therefore, based on text classification algorithms, this paper refines the results by combining adjectives and adverbs in the emotion dictionary and proposes a loss mechanism with temporal information to improve the accuracy of emotion classification.

### 2.3. Research on the Spatial Environment

The built environment carries the activities of individuals, and unlike the natural environment, it includes land use, buildings, transportation, and landscape [50]. Built environment assessment originated from the investigation of satisfaction with the quality of the living environment. With the development of behavioral research, environmental assessment theory was gradually formed [51]. The evaluation elements of the built environment can be roughly divided into five aspects: social environment, land use, road traffic, eco-space, and public service, as shown in Table 1. Mao et al. used eight built environment elements based on streetscape maps to evaluate the street environment, which focused more on assessing the walkability of street spaces; indicators related to road traffic, including pavement, parking occupancy, accessibility, and pedestrian scale; indicators related to eco-space, including sightline and green landscapes [52]. Chen used seven elements to explore the emotional space and built environment in Nanjing, including traffic, residence, public service, etc.; these elements could also be divided into five aspects [53]. Lin et al. studied happiness and the built environment by using four aspects of the built environment: indicators related to the social environment, including population density, deviation index of employment, and residence; indicators related to road traffic, including buses available; indicators related to eco-space, including green rate; indicators related to public service, including diversity of public facilities [54]. Lu used four categories of built environment elements in the study of the emotional health of older environments: indicators related to social environment, including population density; indicators related to land use, including land type and plot ratio; indicators related to road traffic, including bus station density and street height to width ratio; indicators related to eco-space, including view rate and sky openness [55]. Xie et al. analyzed Shanghai’s built environment using three aspects: social environment indicators, including job–housing relationships; land use indicators, including land use intensity, spatial linkages, and nodes; road traffic indicators, including travel behavior [56].

Spatial elements are complex, and the built environment evaluation elements vary when the research aims are different. Different spatial scales, from urban areas to streets, can be used as units of study for the built environment, which can present different spatial characteristics. In accordance with previous studies and considering the availability of data, ten indicators were selected to assess the built environment of Shanghai in five aspects: land use degree, job–housing relationship, road traffic, green rate, and density of five service facilities.

## 3. Materials and Methods

### 3.1. Study Area and Data Sources

The research object was Shanghai, the national central city and a megacity in China. The study area covers 16 districts, with a total area of 6340.5 square kilometers. According to the United Nations World Health Organization’s criteria for age classification, people aged 15–45 are classified as youth. According to the age of Weibo users, the active users are mainly young people. Based on the Weibo data in Shanghai, this study analyzed the emotional characteristics of youth.

Between 19–25 July 2021, by filtering the residence address of users, this study used a Python crawler to obtain 15,675 Weibo comments from Shanghai youth residents. Each Weibo review mainly contained a user ID, posting time, the longitude and latitude of the check-in point, and the review itself (as shown in Table 2).

To ensure the validity of the data and improve the accuracy of sentiment recognition, the original data were screened before being processed. First, duplicate data were excluded; garbled texts caused by the failure of text language parsing were removed in the crawling process. Second, blank texts such as images and videos and advertisements without subjective opinions were removed. Finally, Weibo comments beyond the scope of Shanghai were deleted. While reviewing Weibo comments, pictures and videos could not be accessed due to platform limitations. They were displayed in text as “Weibo pictures” and “Weibo videos” and thus research considered them as meaningless. After this process, there were a total of 10,916 valid reviews. The spatial distribution of Weibo check-in points in Shanghai is shown in Figure 2. Correlating the number of Weibo comments in each district with the resident youth population, the results are highly significant, indicating that the one-week Weibo data used in the study can be a realistic reflection of public sentiment.

### 3.2. Text Sentiment Classification

#### 3.2.1. Dataset for Training and Testing

The dataset used for algorithm training and testing was provided by the Social Computing and Information Retrieval Research Center of the Harbin Institute of Technology. The raw data originated from the Micro-Hotspot Big Data Research Institute of Sina Weibo. The dataset is divided into two parts. The first is the general Weibo dataset, which is randomly obtained Weibo content that is not specific to a particular topic and instead covers a wide range of topics. The second is the COVID-19 dataset, which is a collection of comments obtained during the epidemic period using relevant keyword filtering and whose content is related to COVID-19. Each comment was labeled as one of the following six sentiments: “Happy”, “Surprise”, “Neutral”, “Anger”, “Sad”, and “Fear”. This dataset consisted of 27,768 reviews.

#### 3.2.2. Framework

The sentiment classification algorithm framework was applied to explore the relationship between youth sentiment and the built environment. To obtain the final output for further analysis, the raw text went through three stages: Baseline, Refinement, and Reanalysis. Due to difficulty in annotating the dataset, we could not obtain a more detailed sentiment classification from a Baseline model. In order to calculate sentiment intensity, feeding the sentiment label into the Reanalysis module was used to output the final intensity through some sample-free training methods. The overall algorithm framework is shown in Figure 3.

#### 3.2.3. Pretrained Vocabulary Vector

One of the most important tasks in natural language and emotion recognition is the encoding of a character or word, which loses its contextual information in traditional one-shot encoding. From the classical 2-g to the n-gram, all aim to make the word vector more relevant to the contextual information. Word2vec is a normal tool for acquiring distributed word vectors. After large-scale pre-training of the data, a neural network consisting of a shallow hidden layer and softmax was constructed. Word2vec obtains more contextual information from word vectors, which is important for emotion-recognition tasks. Therefore, Word2vec was used to construct the word vectors in this study.

#### 3.2.4. Baseline Module

Most natural language processing tasks are based on the Seq2seq model, which consists of two parts, the encoder and decoder. After the proposal of the Attention mechanism, the Seq2seq model with Attention was improved for each task. In this paper, the Baseline module of the sentiment recognition adopts the Seq2seq model based on Transformer, which consists of a stack of six encoder and decoder modules, as shown in Figure 4. The encoder module is mainly used to extract features from the word vectors, and the encoder module focuses more on the word-to-word (character-to-character) relationships. The decoder module is mainly used to recode the extracted features, which will make the different tasks more flexible.

“Encoder#1” is an encoder module that consists of a Multi-Head Attention layer and a Fully Connected Feed Forward Networks layer, for converting the input corpus into feature vectors. “Decoder#1” is a decoder module that consists of a Masked Multi-Head Attention layer, a Multi-Head Attention layer, and a Fully Connected Feed Forward Networks layer, for outputting the final probabilities of sentiment labels.

#### 3.2.5. Refinement Module

Multi-Loss Constraint

The loss function is often used to describe the convergence of a model. This study built three transformer modules to obtain semantic information at different depths. The first transformer module obtains information about subject-verb-object, the second obtains information about the connection of subject-verb-object sequences, and the third obtains the meaning of words with depth. The shallow information of the first layer is then stitched into the deep information of the next two layers, as shown Figure 5, and both the latter two layers are passed through the loss constraint. With the multi-loss constraint, the model obtains better emotion capture. In general, this study aimed to design a parallel loss structure like RNN, which superimposes semantic information at the encoder layer. This structure improved the overall recognition of emotion by superimposing continuous semantics on the output feature map.

2.Data Augmentation

When training a machine learning model, it is necessary to map an input of a picture or text to a corresponding output. The optimization goal pursues the optimal point of model loss. However, when tuning the model using the correct approach, it turns out that a model with higher prediction accuracy usually contains millions of floating-point parameters, and training these parameters often requires more samples to ensure correctness.

In order to increase the amount of data in the training sample, data augmentation must be used. This means data is augmented with a certain amount of extra data without concern for data annotation. This study used two data augmentation methods. The first is back translation (translating twice, e.g., Chinese to English and then English to Chinese), and the second is easy data augmentation: synonym replacement, insertion, exchange, and deletion. Data augmentation can improve the robustness of the model.

#### 3.2.6. Reanalysis Module

In order to be able to describe sentiments in more detail, the Reanalysis Module calculates the sentiment intensity. In fact, for fine-grain evaluation, metrics need to be performed with a larger and more precise dataset. With the existing data, this study used the sentiment dictionary HowNet to extract adjectives and adverbs in comments for a more detailed analysis of sentiment. The reanalysis module is shown as Figure 6.

The module mainly consisted of a shallow deep learning network for regression and a fusion module incorporating a sentiment dictionary. By analyzing the adjectives and adverbs before and after the sentiment words in the text against the sentiment dictionary, the sentiment intensity of the text was determined. The adjectives and adverbs were encoded as word vectors, and the encoded information was fused into the final regression task to obtain the results.

### 3.3. Built Environment

Based on the previous review of international studies on the built environment, it was found that indicators for evaluating the built environment can be divided into five aspects; this study decided to select ten elements from these five aspects to evaluate the built environment according to the availability of data.

#### 3.3.1. Land Use Degree

Land use is an important influencing factor of the built environment, which can reflect the spatial layout of urban functions, the diversity of land use in a particular range, and the level of economic development and irrationality in construction. In this study, land use degree, based on the entropy index, was used to measure the land use of each research unit in Shanghai, and the formula is as follows:(1)$$MixUsed=-\frac{\sum_{i=1}^{N}p_{i}\ln p_{i}}{\ln N}$$
where $p_{i}$ is the ratio of the area of each type of land to the total area of the unit; N is the land type.

#### 3.3.2. Job–Housing Relationship

Smartphones record the location trajectory of users at certain time intervals; from this, the residence and workplace of users can be identified. In this study, the distribution of the residential population and employment population of Shanghai youth was extracted from mobile phone signaling data to analyze the job–housing relationship. The job–housing index is important for evaluating whether the distribution of urban residential areas and jobs is reasonably allocated; it is an important element of the built environment. The job–housing index is expressed by the ratio of the youth resident population in each district to the employed youth population.

#### 3.3.3. Road Traffic

Road traffic includes two built environment elements: density of road network and density of public transportation stations. Road network density measures the connectivity of the regional transportation network and reflects the convenience of transportation in the region, which is conducive to people’s travel activities. At the same time, the higher the road network density, the greater the traffic volume in the area. The density of road networks in this study was calculated as the ratio of the total length of the road network in the district to the area of the district. Public transportation stations include bus stations and subway stations, and this study used the ratio of the number of public transportation stations in the district to the area of the district to indicate the density of public transportation stations. This in turn was used to reflect the convenience of public transportation in the region.

#### 3.3.4. Green Rate

Green space is not only an important place for public activities but also a major part of determining environment quality. The green rate is expressed as the ratio of the green space area in the district to the area of the district. The ratio of green space reflects the distribution of green space and the built environment.

#### 3.3.5. Service Facilities

Five categories of public service facility were selected: shopping facilities, food services, entertainment facilities, medical services, and exercise facilities. The ratio of the number of shopping facilities, food services, entertainment facilities, medical services, and exercise facilities to the area of the district was used in the study.

## 4. Results

### 4.1. Text Sentiment Analysis

The results of sentiment classification were obtained after the Baseline Module and Refinement Module of Weibo texts. The accuracy of our model classification was compared with popular models, and the results are shown in Table 3.

The results of the sentiment classification distribution of the dataset are shown in Figure 7, and the detailed results for each district are shown in Table 4.

Based on Table 4, the number of texts labeled “Neutral” was the highest, accounting for about 37% of the total sample. Comments with “Neutral” labels do not have an emotional character. The number of texts labeled “Fear” was the lowest (only nine); thus, it is not representative of prominent sentiments. In subsequent studies, the other four sentiments will be focused on. In general, when only “Happy”, “Surprise”, “Sad”, and “Angry” were considered, Shanghai youth had 45% positive sentiments (“Happy” and “Surprise”) and 55% negative sentiments (“Sad” and “Angry”), with more negative sentiments than positive sentiments in the one week. From the results of each district, the district with the highest percentage of positive sentiment was Qingpu (about 52%), followed by Fengxian (about 50%), as shown in Figure 8. Qingpu, in recent years, has converged two national strategies: a demonstration green zone and integrated ecological development of the Yangtze River Delta and the Hongqiao International Hub. These strategies have allowed Qingpu to continue to upgrade its core urban functions and service quality. From the current results, only these two districts have a positive-polarity sentiment. Qingpu and Fengxian both belong to the construction area of “The Five New Cities in Shanghai”, which highlights the quality of space, in accordance with standards and levels applicable beyond the construction of the central city.

The “Happy” label accounted for the highest proportion of texts in Chongming, about 31%. Chongming is a very popular tourist destination in Shanghai, with a beautiful ecological environment and a higher sense of well-being. The “Sad” label accounted for the highest number of texts in Jinshan, about 28%, and it also was responsible for the highest proportion of negative sentiment. Jinshan is located in the southwest of Shanghai and is an industrial area. Although there are abundant land resources, its built environment is poor, and the construction of public service facilities needs to be improved.

The intensity of the five sentiments is shown in Table 5. There are five values of sentiment intensity: −2, −1, 0, 1, and 2. The value “2” indicates greater intensity and the value “−2” indicates less intensity. Taking the “Happy” sentiment as an example, “2” means very happy, and “−2” means happy. Overall, the intensity of all types of sentiment is mainly at a high level. The average intensity of “Happy” in each district is relatively similar; the average intensity of “Surprise” is significantly different, with the highest intensity in Songjiang (average intensity of 1.17) and the lowest in Chongming (average intensity of 0). “Anger” is the highest intensity among all types of sentiments, and the intensity of each district is relatively similar; the average intensity of “Sad” in each district is at a lower level. Districts with higher intensity of positive sentiments are usually accompanied by higher intensity of negative sentiments as well. In addition to the central area, “The Five New Cities” also have higher sentiment intensity. This result suggests that paying more attention to the spatial environment can enhance the positive emotions of youth.

### 4.2. Built Environment Analysis

Based on the current land use data in Shanghai in 2014, this study extracts five types of land use, including residential land, commercial land, industrial land, green land, and water area. When calculating the job–housing index for youth, the resident and employed populations of each unit in Shanghai were screened by mobile phone signaling data. The road network data used in this study are obtained from OpenStreetMap road data, which are divided into express roads and arterial roads, among which express roads include highways and urban expressways, and arterial roads include urban main roads and branch roads. For transport stations and services, we used the geocoding service based on Web API provided by the Baidu Map to extract Shanghai bus stops, subway stations, and five types of public service facility points.

The statistical results of all elements of the built environment are shown in Table 6. Land use degree varies by region. Pudong has the highest land use degree, while Huangpu has the lowest. Pudong is the largest district in Shanghai and also has the largest resident population. Pudong is a center of economy and technology, so it has a high land use degree. From the perspective of the job–housing relationship, Huangpu has a high job–housing relationship index, and as the core area of Shanghai, it has a high density of young people, but the distribution of jobs is low. Baoshan and Songjiang are the areas with the lowest job–housing relationship index, providing relatively more jobs for young people. Road traffic conditions are better in urban centers and worse in suburban areas. The green rate is higher in suburban areas and lower in urban centers; Chongming has a green rate of 62%. The distribution of daily service facilities is similar to road traffic and is mainly concentrated in Huangpu, Jingan, Changning, and other areas.

### 4.3. Correlation Analysis

The correlation analysis between the intensity of sentiment and the built environment was performed by SPSS at the scale of districts, and the results are shown in Table 7. Only the job–housing relationship index and green rate were significantly correlated with the intensity of “Surprise”.

At the scale of 500 m by 500 m, the correlation between sentiment intensity and built environment was also analyzed with SPSS, and the results are shown in Table 8. Land use degree and green rate were significantly correlated with the intensity of “Happy” and “Surprise”; road traffic, green rate, and service facilities were negatively correlated with the intensity of “Happy” and “Surprise.” Land use degree and green rate were negatively correlated with the intensity of “Sad”, “Neutral”, and “Angry”. The job–housing relationship, road traffic, and service facilities were significantly correlated with the intensity of “Sad” and “Neutral”; road traffic and service facilities were significantly correlated with the intensity of “Anger”. For land use degree, the larger the value, the greater the intensity of the “Happy” and “Surprise” sentiment. The greater the land use degree, the higher the diversity of facilities; the better the living environment, the more it can stimulate positive sentiment. For the job–housing relationship, the greater the value, the more residential the area and the lower the intensity of “Happy” and “Surprise”. For road traffic, road-network-dense areas with better accessibility stimulated the intensity of negative sentiments and reduced the intensity of positive sentiments. Regarding service facilities, the higher the density of service facilities in the area, the greater the intensity of negative sentiment, and the lesser the intensity of “Happy” and “Surprise”. Among all the built environment elements that had a high correlation with sentiment intensity of the youth, the density of food services had the greatest impact on all types of sentiment.

As the result, the built environment elements extracted from the scale of 500 m by 500 m are more correlated with sentiment intensity than district scale. This is because the scale of 500 m by 500 m more accurately reflects the actual range of individuals’ activities and more realistically reflects the built environment conditions that people are exposed to in their daily activities.

## 5. Discussion

To our knowledge, this is the first study to use Weibo comments in Shanghai to assess built environment quality and sentiment through a machine learning algorithm based on an attention mechanism. This study measured the sentiment characteristics of urban youth in Shanghai, reflecting on how the built environment affects sentiments on different scales. This study represents a critical first step in developing technology for harnessing social media data using a novel data source to monitor changes in public sentiment. Our findings indicate that social media data provide a more realistic picture of the sentiment characteristics of urban youth and that elements of the built environment at smaller scales have a significant effect on the intensity of all types of sentiments.

In terms of the spatial form of the city, land use degree should be appropriately increased, as land use degree is positively correlated with “Happy” and “Surprise” sentiments. Increasing land use degree can promote the expression of positive sentiments and reduce the proportion of negative sentiments. By observing land type in the study unit, it was found that commercial land was able to generate more positive sentiments. This indicates that the youth group is highly dependent on commercial facilities in their daily lives and that abundant resources of commercial facilities have a positive effect on the sentiments and well-being of youth. This can strengthen the mixed use of land, improve functional diversity, and increase the vitality of an area.

For the job–housing relationship, it is important to promote a balance of jobs and housing. The larger the job–housing relationship index is, the more separated jobs and housing are. Separation of jobs and housing will make the daily commute distance for youth longer, causing traffic congestion and waste of resources and therefore a higher intensity of the “Sad” sentiment. The job–housing relationship for youth can reflect the distribution pattern of youth commuting activities. The youth group is the main group of commuters, and if the job–housing relationship is improved, it will not only reduce the traffic congestion in the city, but also improve the employment attractiveness of the region. In areas where jobs are clustered, living areas and supporting living service facilities should be planned in order to better optimize spatial structure.

For road traffic, it is important to focus on the negative impacts of high accessibility because there is a high demand for traveling for urban youth. The higher the density of road networks and public transportation stations, the better the accessibility of the area and the more convenient it is for travel. However, high accessibility can cause problems such as excessive traffic, noise, and air pollution and can degrade environmental quality. Therefore, to avoid negative sentiment, it is important to pay attention to the quality of services and any traffic problems.

From the perspective of green space planning, the green rate of the central city should be increased. Many scholars have experimentally demonstrated that green natural resources can alleviate negative sentiments and enhance positive sentiments, and this is no exception for youth. The higher the green rate, the higher the intensity of positive sentiment. In urban planning, we should focus on the construction of various types of parks and green spaces and increase the density and coverage of green spaces to create positive emotional space.

For service facilities, attention needs to be paid to distribution. The distribution of service facilities is related to land use. Because of the high demand for food services among youth, the density of food services has a higher impact on sentiment intensity compared to other services. The layout of daily service facilities affects the basic quality of life of the youth; higher density of various types of service facilities, on the contrary, promotes negative sentiments. The layout of service facilities should avoid misallocation and waste of resources. At the same time, higher quality services should be provided to create a positive emotional environment.

## 6. Conclusions

Taking Shanghai as an example, this study explores the relationship between youth sentiment and the built environment by using attention-based machine learning methods. Shanghai is building a “Global City of Excellence”, and using it as an example can provide experience and recommendations for other metropolitan areas. According to this study, the sentiment of Shanghai youth tends to be negative, with a high percentage of “Sad” and “Anger” labels. Industrial areas with poorer built environments have a higher proportion of negative sentiments, while “The Five New Cities in Shanghai” have a higher proportion of positive sentiments. The sentiment intensity of Shanghai youth is at a higher level. Areas with higher intensity of positive sentiments are usually also accompanied by higher intensity of negative sentiments. At a smaller scale, built-environment elements have a significant effect on youth sentiments, and improving the built environment can promote positive sentiments and enhance the well-being of youth.

## Figures and Tables

**Figure 1 ijerph-19-04794-f001:**
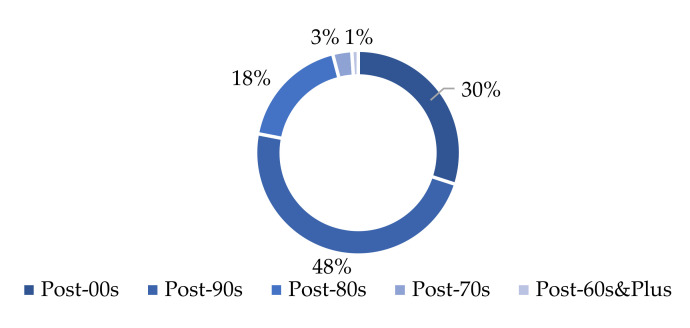
Age distribution of Weibo users.

**Figure 2 ijerph-19-04794-f002:**
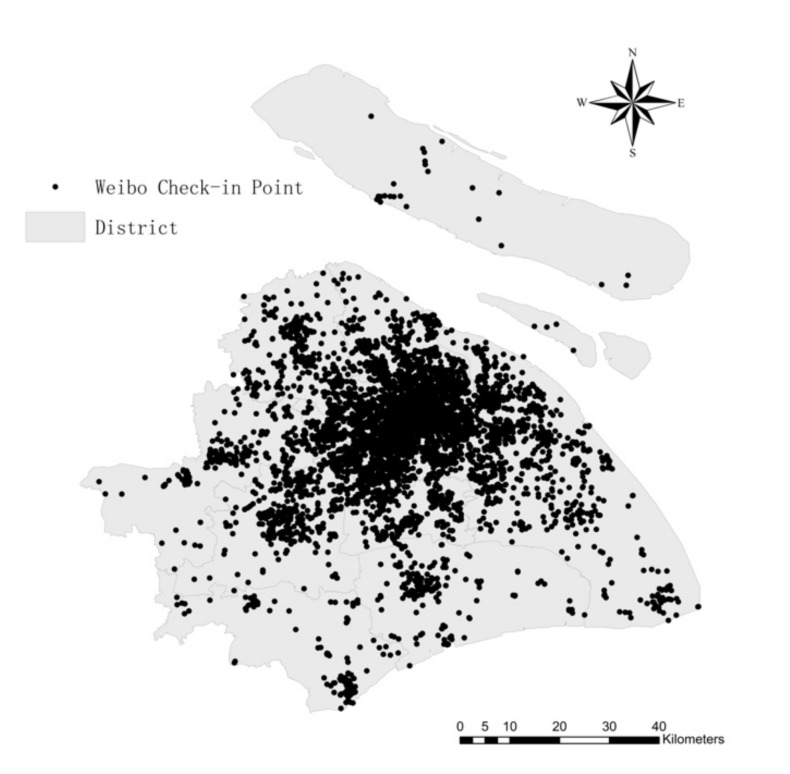
Distribution of Weibo check-in points in Shanghai.

**Figure 3 ijerph-19-04794-f003:**
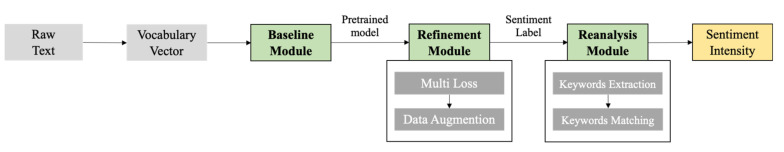
The sentiment classification algorithm framework.

**Figure 4 ijerph-19-04794-f004:**
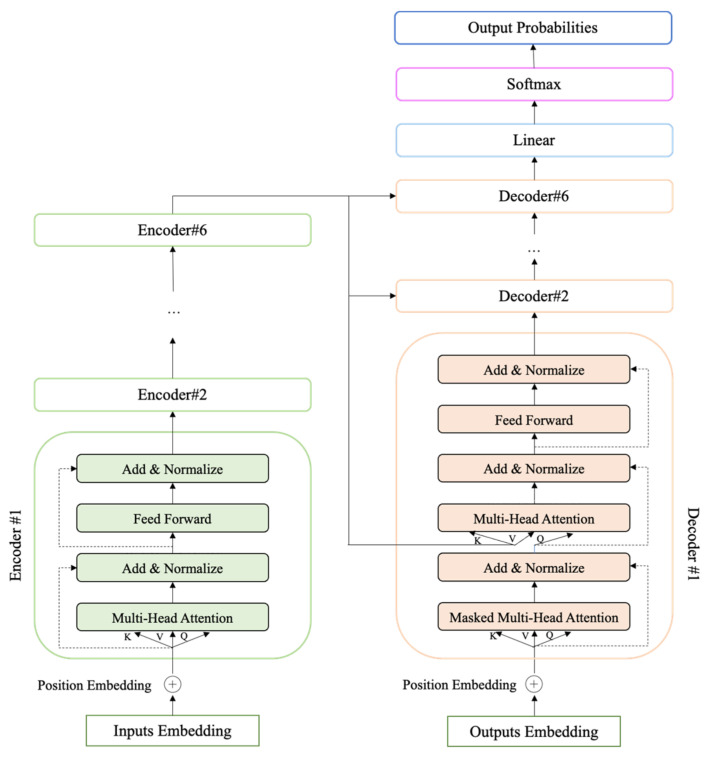
Baseline module [64].

**Figure 5 ijerph-19-04794-f005:**
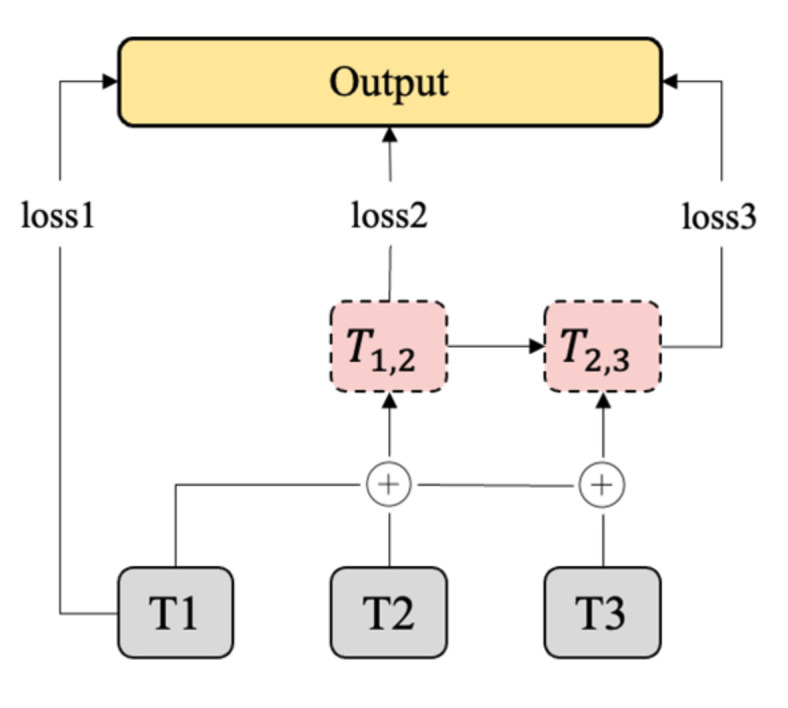
Multi-loss constraint framework.

**Figure 6 ijerph-19-04794-f006:**
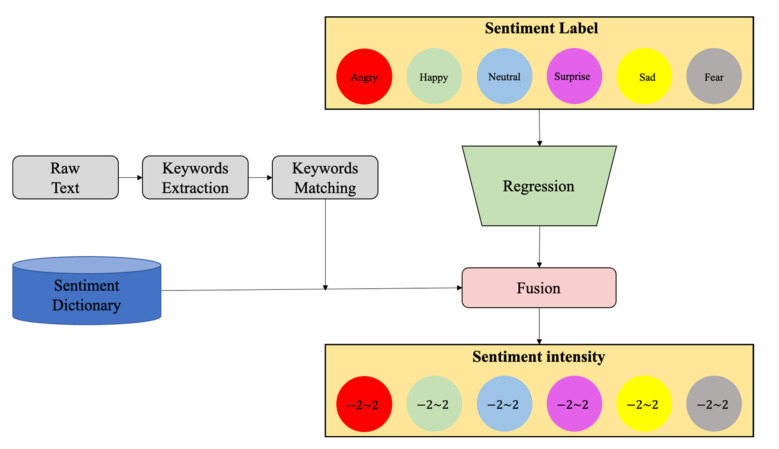
Reanalysis module.

**Figure 7 ijerph-19-04794-f007:**
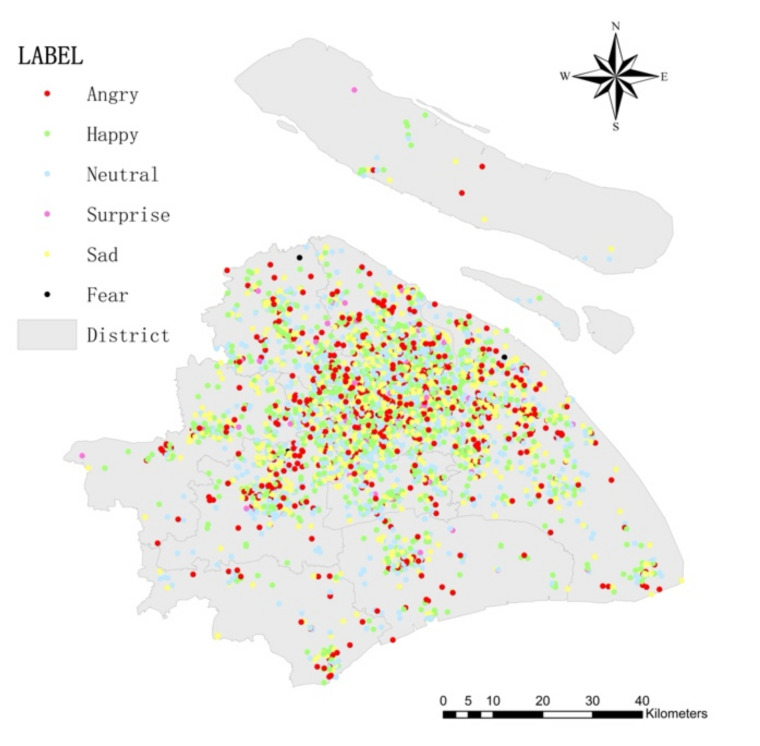
Sentiment classification distribution.

**Figure 8 ijerph-19-04794-f008:**
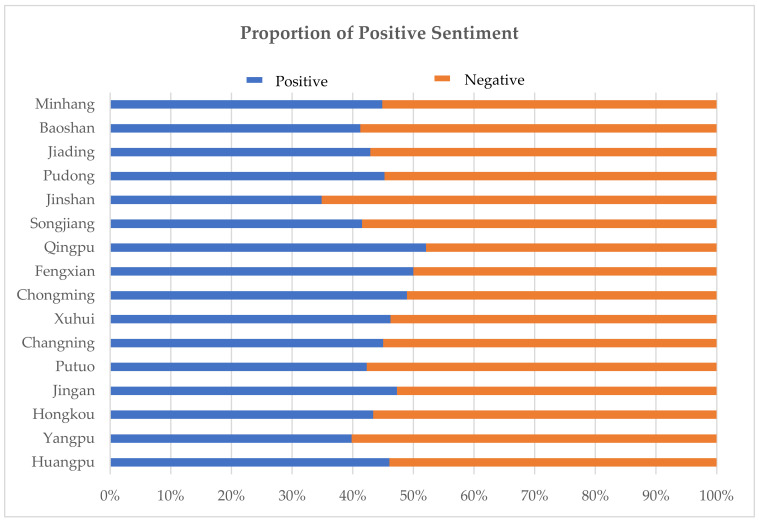
Proportion of positive sentiment.

**Table 1 ijerph-19-04794-t001:** Summary of previous studies.

Researchers	Built Environment Evaluation Elements				
	Social Env.	Land Use	Road Traffic	Eco-Space	Public Service
Lin et al. [54]	Population density, Deviation index of employment and residence		Buses available	Green rate	Diversity of public facilities
Lv et al. [55]	Population density	Urban spatial structure	Bus station density, Street height to width ratio	Green view rate Sky openness	
Xie et al. [56]	Job–housing relationship	Land use intensity	Residents’ travel behavior		
Xu et al. [57]				Soundscape	
Long et al. [58]			Street crossing facilities, Motor vehicle and non-motor vehicle isolation, Walkway width	Street greening	Street facilities
Leslie [59]		Land use	Traffic safety, Street connectivity, Traffic flow	Green rate	Infrastructure
Ettema [60]	Attractiveness		Accessibility, Traffic safety		Facilities
Ewing [61]		Land use	Accessibility	Green rate	
Yuan [62]	Population density	Land use degree, Land type, Residential land ratio	Bus station density, Density of road network, Intersection density	Green rate	
Wang et al. [63]		Land use degree	Density of road network, Buses available		Density of public service

**Table 2 ijerph-19-04794-t002:** Example of Weibo comments.

ID	Time	Longitude	Latitude	Weibo Comments
1	19 July 13:43:43	121.4861	31.23672	Chinese Comments: ‘好久不更博,最近把微博给忘了’. English Comments: ‘I forgot about Weibo recently, it’s been a while since I updated’.
2	19 July 17:21:14	121.4422	31.22382	Chinese Comments: ‘失踪人口回归’. English Comments: ‘Return of missing persons’.
3	19 July 00:18:53	121.4446	31.22577	Chinese Comments: ‘吃六个带两个回家,上海限定豫园奶昔也太好喝’. English Comments: ‘Eat six and take two home, Shanghai limited Yuyuan milkshakes are too good’.

**Table 3 ijerph-19-04794-t003:** Model evaluation of sentiment analysis.

Method	Score (Acc %)
FastText	0.639
TextCNN	0.657
TextRCNN	0.645
Transformer	0.650
* **Ours** *	* **0.697** *

**Table 4 ijerph-19-04794-t004:** Results of sentiment classification.

District	Sentiment Label						
	Happy	Surprise	Neutral	Angry	Sad	Fear	Total
Huangpu	380	32	649	182	301	0	1544
Yangpu	84	12	145	62	83	0	386
Hongkou	77	8	137	41	70	0	333
Jingan	222	13	292	107	155	0	789
Putuo	107	3	136	67	83	1	397
Changning	131	9	166	58	113	0	477
Xuhui	245	14	314	134	167	2	876
Chongming	22	1	25	14	10	0	72
Fengxian	54	4	67	27	31	0	183
Qingpu	154	8	252	64	85	0	563
Songjiang	154	6	200	94	131	1	586
Jinshan	29	1	34	23	33	0	120
Pudong	684	48	900	370	516	4	2522
Jiading	131	8	131	74	111	1	456
Baoshan	122	10	133	80	108	0	453
Minhang	310	22	419	152	256	0	1159
Total	2906	199	4000	1549	2253	9	10,916

**Table 5 ijerph-19-04794-t005:** Sentiment intensity.

District	Happy		Surprise		Angry		Sad	
	Mean	SD	Mean	SD	Mean	SD	Mean	SD
Huangpu	0.78	0.45	−0.09	0.90	1.19	0.97	−0.58	0.90
Yangpu	0.71	0.44	−0.17	0.31	1.06	0.90	−0.64	0.74
Hongkou	0.78	0.51	−0.38	0.23	1.27	0.73	−0.61	1.09
Jingan	0.70	0.52	−0.35	0.40	1.24	0.80	−0.54	0.93
Putuo	0.71	0.39	1.00	0.67	1.06	0.83	−0.45	0.92
Changning	0.71	0.43	−0.11	0.77	1.10	0.85	−0.55	0.96
Xuhui	0.74	0.43	−0.07	0.78	1.15	0.87	−0.52	1.02
Minhang	0.66	0.53	0.09	0.72	1.33	0.88	−0.67	0.89
Fengxian	0.80	0.64	−0.50	0.25	1.19	0.97	−0.77	0.50
Qingpu	0.75	0.43	−0.13	0.61	1.09	0.99	−0.42	1.16
Songjiang	0.69	0.54	1.17	0.47	1.35	0.55	−0.73	0.86
Jinshan	0.72	0.48	1.00	0.00	1.26	0.63	−0.33	1.19
Pudong	0.69	0.54	−0.31	0.67	1.19	0.81	−0.65	0.87
Jiading	0.77	0.50	−0.50	0.50	1.05	0.92	−0.46	1.08
Baoshan	0.81	0.46	−0.10	0.09	1.14	0.82	−0.65	0.86
Chongming	0.73	0.38	0.00	0.00	0.93	1.07	−0.30	1.01

**Table 6 ijerph-19-04794-t006:** Built environment elements.

District	Land Use Degree	Job–Housing Relationship	Road Network (km/km^2^)	Transportation Station	Green Rate	Shopping Facilities	Food Services	Entertainment Facilities	Medical Services	Exercise Facilities
Huangpu	0.52	1.32	17.86	1.32	0.14	694.03	230.7	29.7	44.85	25.85
Yangpu	0.65	0.56	13.77	0.45	0.11	207.49	99.56	14.02	23.29	11.69
Hongkou	0.53	0.67	17.05	0.77	0.1	362.53	159	20.34	39.36	20.08
Jingan	0.67	1.01	16.55	0.71	0.12	508.99	183.57	21.09	40.19	21.28
Putuo	0.71	0.61	15.76	0.4	0.24	243.69	96.37	13.12	20.85	12.31
Changning	0.68	0.90	16.81	0.48	0.31	248.88	122.44	16.64	32.07	17.26
Xuhui	0.63	1.07	16.27	0.63	0.22	232.15	114.84	14.79	24.8	17.09
Minhang	0.74	0.51	6.73	0.27	0.21	66.06	31.01	4.36	4.65	3.53
Fengxian	0.76	0.37	3.94	0.15	0.47	21.1	6.25	1.11	1.38	0.47
Qingpu	0.74	0.40	3.88	0.15	0.41	19.15	6.77	0.76	1.15	0.54
Songjiang	0.74	0.35	5.15	0.17	0.42	30.36	13.94	2.05	2.03	1.22
Jinshan	0.64	0.43	4.04	0.21	0.54	15.28	4.74	0.86	1.1	0.35
Pudong	0.84	0.53	6.08	0.18	0.31	44.35	18.49	2.37	3.46	2.05
Jiading	0.77	0.41	5.8	0.13	0.33	41.49	16.15	1.95	2.74	1.32
Baoshan	0.78	0.30	6.67	0.13	0.55	72.46	27.9	3.8	4.69	2.64
Chongming	0.77	0.41	2.9	0.13	0.62	4.04	0.78	0.49	0.34	0.1

**Table 7 ijerph-19-04794-t007:** Correlation between public sentiment intensity and built environment (scale: district).

Built Environment	Happy	Surprise	Angry	Sad
Land Use Degree	0.207	−0.143	0.132	−0.238
Job–housing	0.261	−0.590 *	−0.106	−0.135
Road Network	−0.292	0.462	−0.036	0.016
Transportation	−0.168	0.436	0.061	0.042
Green Rate	−0.013	−0.559 *	0.03	0.051
Shopping Facilities	−0.162	0.402	0.055	−0.003
Food Services	−0.198	0.413	0.024	0.013
Entertainment	−0.225	0.424	0.033	−0.003
Medical Services	−0.205	0.37	−0.003	0.034
Exercise Facilities	−0.234	0.436	0.016	0.04

* Significant at the 0.05 level.

**Table 8 ijerph-19-04794-t008:** Correlation between public sentiment intensity and built environment (scale: 500 m × 500 m).

Built Environment	Happy	Surprise	Angry	Sad
Land Use Degree	0.014 *	0.076 **	−0.034 **	−0.052 **
Job–housing	−0.003	−0.012	0.007	0.021 **
Road Network	−0.044 **	−0.212 **	0.102 **	0.166 **
Transportation	−0.003	−0.103 **	0.034 **	0.074 **
Green Rate	0.018 **	0.161 **	−0.078 **	−0.131 **
Shopping Facilities	−0.052 **	−0.217 **	0.090 **	0.150 **
Food Services	−0.080 **	−0.282 **	0.107 **	0.221 **
Entertainment	−0.064 **	−0.242 **	0.106 **	0.194 **
Medical Services	−0.071 **	−0.197 **	0.085 **	0.150 **
Exercise Facilities	−0.053 **	−0.268 **	0.117 **	0.205 **

** Significant at the 0.01 level, * significant at the 0.05 level.

## Data Availability

Not applicable.

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
