# Peer review of "Exploring the Relationship between Urban Youth Sentiment and the Built Environment Using Machine Learning and Weibo Comments"

_ijerph, 2022, doi:10.3390/ijerph19084794_

Round 1
Reviewer 1 Report
This research is well conducted and written. I enjoy reading this paper and believe it to raise the importance of machine learning algorithm approach to the relationship between the build environment and urban youth sentiment in the context of emotion geography. Moreover, the concepts in the paper are well presented in response to practical implications of the spatial form of the city. Very well done!!
Author Response
Thank you for reviewing this article and thank you for your kind words. I will revise the article and improve the details, hoping to better explain the importance relationship between the built environment and youth sentiment.
Reviewer 2 Report
This manuscript explores the relationship between youth sentiment and various built environment elements in Shanghai, and the accuracy of sentiment classification is improved by modifying the model. However, some revisions are still needed. Provided the following comments and suggestions:
- The research innovation of this manuscript should be more prominent.
- This manuscript still needs a thorough and criticism-featured Literature Review, rather than describing the research findings of other scholars. The existing literature review is insufficient.
- In the Introduction section, the author wrote “According to the "Weibo User Development Report in 2020", those people born after the 1980s accounted for 96% of all active Weibo users.” and I doubt whether it is appropriate to use the word “urban youth” in the title of this manuscript. Moreover, the specific relationship between urban youth and the built environment is ignored in the following analysis.
- Lack of the section of Discussion. It is necessary to carry out an in-depth analysis of the three parts of the results, such as the reason for the spatial heterogeneity of emotional intensity and elements of the built environment, as well as causes of the correlation between the two.
- What new global knowledge can this paper contribute to the existing international literature? How to link the findings and conclusions in this paper with the previous findings and conclusions from other countries? Its introduction, analysis, and discussions should be beyond the local case itself. This can help the paper to attract more international readers. However, there are still deficiencies in the contribution of global knowledge.
- The structure of this manuscript is loose and chaotic, and the research focus is not prominent enough.
- The font in the legend of Figure 2 is difficult to be recognized and needs to be further modified.
Author Response
Thank you for reviewing this article and your valuable suggestions. Your suggestions are very helpful for me to improve the quality of my article. I will improve the following contents according to your suggestions:
- Highlight research innovations;
- Increasing research Gap;
- Increasing the description and definition of research subject to enhance global interpretability;
- More in-depth discussion of the results;
- Adjust the structure of the article and improve the details.
Thank you again for your comments and suggestions. I hope this article can better explain the importance of the relationship between the built environment and youth sentiment with your help.
Reviewer 3 Report
Title: Exploring the Relationship Between Urban Youth Sentiment and the Built Environment by Using Machine Learning and Weibo Comments
Overview and general recommendation
Actually, I found the paper to be overall well written and much of it to be well described. Using a machine learning algorithm of attention mechanism, this study calculates the sentiment label and sentiment intensity of each comment. Correlations were assessed between the sentiment intensity and built-environment factors.
However, I found some description of very important points were inadequate or completely missing. Therefore, I recommend that a major revision is warranted and ask the authors to specifically address each of my comments in their response.
Major comments:
1)Since the object of this paper is the relationship between urban youth sentiment and built environment, it seems that there is no clear definition about urban youth, whether permanent residents or transients? And is there any explanation of urban youth age?
2)Page 6, line 260-261, Actually, in literature review, authors have mentioned that emotions were divided into seven types in previous studies, but in this paper, the sentiments were classified into six types, why is the ‘evil’ excluded?
3)Page 6, Since authors have analyzed and summarized the previous research, there are many built environment evaluation elements. How does authors select indicators? It should be given support.
4)Page 6, line270-271, The paper chose 157675 Weibo reviews from Shanghai Weibo points of interest, Does the author consider the people who traveled to Shanghai or went on short-term business trips during this period, lacking good understanding of the built environment in Shanghai? Does the author regard they as the part of urban youth?
5)Line275-280 It seems that there is no clear description about the specific standard of reviews check. How the authors judge duplicate reviews? And concerning the topic that images and videos have no subjective opinions, is it too absolute?
6)It can be seen from the Figure 7 that there are few sentiment data in Chongming district. Is it unconvincing compared with the data in other districts?
7)Page 13, There is no basic for statistical results of all elements of the built environment in Table 6, so it is suggested to supplement data sources during the survey.
8)Line457-459 “Only the job-housing relationship index and green rate were significantly correlated with the intensity of Surprise. Correlation analysis is not meaningful at this scale.” Why factors are not significantly correlated at the scale of districts? Is there any persuasive explanation?
Author Response
Thank you for reviewing this article and your valuable suggestions. Your suggestions are very helpful for me to improve the quality of my article. Here are the answers to some of the questions, and I will improve the following contents according to your suggestions:
Q1, Q4,Q5: During the checking of Weibo check-in data, we choose the users who live in Shanghai through their personal information, so the article described the permanent resident population of Shanghai. Of course, there will be uncontrollable factors, such as the fake phenomenon of information filling. In addition, when we captured Weibo reviews, we could not get pictures and videos due to platform limitations. They would be displayed in the text of "Weibo pictures" and "Weibo videos", so we considered them as meaningless. In addition, some data are garbled, and the screening process mainly removes such meaningless text.
I omitted this key description in my article, I will increasing the description and definition of urban youth to enhance global interpretability;
Q2: Due to the limitation of the dataset for training and testing, we are not able to do more recognition of sentiment types, which will be improved in the future.
Q3, Q7: The selection of built environment elements is also limited by data sources. Through a literature review, we selected elements for which data are currently available and have been shown to have a relationship with sentiment. And We will supplement the data sources in the revised draft
Q6: Chongming has a total of 72 data, and we associated it with the resident population. According to the result, so we think it is convincing. And this question can be reconsidered based on your suggestion.
Q8: We think that this is because at the scale of district, the data cannot reflect sentiment characteristics after homogenization. Maybe We'll do more in-depth discussion of this.
Thank you again for your comments and suggestions. I hope this article can better explain the importance of the relationship between the built environment and youth sentiment with your help.
Reviewer 4 Report
1) This paper appears to be an interesting contribution to research on ‘the relationship between urban youth sentiments and the built environment’;
2) Its academic and scientific values make it suitable for the open access, peer-reviewed International Journal of Environmental Research and Public Health (ISSN 1660-4601);
3) However, there are some minor spell checking, formatting and writing issues that must be addressed by the authors. See, for instance, the “examples” listed below (underlined text):
- Lines 56-58, CHECK TEXT: ‘scholars have also realized that space is not a cold and geometric???; it also carries a wealth of people's sentiments.’;
- Lines 95-97, CHECK and IMPROVE TEXT: ‘By analyzing the distribution characteristics of urban youth sentiments and studying the relationship between sentiments and the built environment in Shanghai, a people-centered strategy to improve the quality of the built environment.????’;
- Line 118, CHECK ENGLISH: ‘positive emotions mainly concentrated in entertainment, sports, food three aspects ???[13].’;
- Table 1., No references to this Table have been found in the text;
- Table 2., PAY ATTENTION TO NON-ENGLISH CHARACTERS; ALSO CHECK THE ‘Built environment evaluation elements’;
- Lines 338-339, CHECK ENGLISH: ‘In general, this study aimed to designed??? its parallel loss structure like a RNN’;
- Lines 378-379, CHECK TEXT: ‘pi is the ratio of the area of each type of land to the total area of the district; N is the type of land type???’;
- Figure 7., PAY ATTENTION TO NON-ENGLISH CHARACTERS;
- Line 434, CHECK for a possible EXTRA ‘(‘ : ‘???(Taking “Happy” sentiment as an example, “2” means very happy’.
Author Response
Thank you for reviewing this article and thank you for your valuable suggestions. Your suggestions are very helpful for me to improve the language of my article, and I will revise the issues you pointed out and check the details of the article again.
Round 2
Reviewer 2 Report
This paper presents the relationship between urban youth sentiment and the built environment by using machine learning and Weibo comments. It can be seen that the author has made a lot of revisions and done a lot of work in response to the previous review, which deserves to be acknowledged.
Author Response
Dear reviewer:
Thank you for reviewing this article and we will check the details of the article again. We hope the changes we have made in the revised manuscript satisfy your concerns.
Best regards
The authors
Reviewer 3 Report
I recommend that a minor revision is warranted and ask the authors to specifically address each of my comments in their response.
1)Based on the fact that people born after the 1980s accounted for 96% of all active Weibo users and the definition of youth people, I agree with the opinion that the active users are mainly young people. However, since author could choose the users who live in Shanghai through their personal information, why not screen data excluding people whose age under 15 or over 45 further through their personal information.
2)line 284-285, “Based on the availability of data, this study will select built-environment elements from the above studies that may have an impact on sentiments.” Revised manuscript still stays at the stage of listing built-environment elements. The concerning literature review (line265-line283) can’t get the conclusion that built-environment elements selected have an impact on sentiments.
3) There are some writing mistakes such as the lack of space between words and the wrong serial number of the literature, which are suggested to be revised.

Author Response
Dear reviewer:
Thank you very much for all the comments. Based on your helpful comments, we have revised our manuscript accordingly. Our responses to the comments are listed below.
Point 1: Based on the fact that people born after the 1980s accounted for 96% of all active Weibo users and the definition of youth people, I agree with the opinion that the active users are mainly young people. However, since author could choose the users who live in Shanghai through their personal information, why not screen data excluding people whose age under 15 or over 45 further through their personal information.
Response 1: Thank you very much for your precious suggestion. In fact, there is a limit to the Weibo data we can access, and information as personal as age is not available. We can filter users living in Shanghai because the address will only be given to the district level scale, and more detailed information is not available. According to the age of Weibo users, the active users born after the 1980s accounted for 96%, so we think it can represent youth aged15-45. Similarly, when calculating the job-housing relationship, we used the age range 15-45 to filter the youth, because the mobile phone signaling data has detailed age information
Point 2: line 284-285, “Based on the availability of data, this study will select built-environment elements from the above studies that may have an impact on sentiments.” Revised manuscript still stays at the stage of listing built-environment elements. The concerning literature review (line265-line283) can’t get the conclusion that built-environment elements selected have an impact on sentiments.
Response 2: Based on the previous review of international studies on built environment, it is found that indicators for evaluating built environment can be divided into five aspects: social environment, land use, road traffic, eco-space, and public service, this study decides to select ten elements from these five aspects to evaluate built environment according to the availability of data. We will modify our paper to express the relationship more specifically.
Point 3: There are some writing mistakes such as the lack of space between words and the wrong serial number of the literature, which are suggested to be revised.
Response 3: We will revise the issues you pointed out and check the details of the article again.
Again, we thank you for your very helpful comments, which we believe improved the quality of this paper. We hope the changes we have made in the revised manuscript satisfy your concerns. If not, however, we are happy to work further to make the paper acceptable for publication.
Best regards
The authors